# Towards Generalizable LLM Multi-Agent System: Identifying Collective Intelligence Factor in LLM Agent Groups

## Abstract

Large language models (LLM)-based multi-agent systems (MAS) have shown impressive performance in solving a wide range of complex problems. However, previous studies mainly focus on designing customized MAS for specific tasks, while a critical research problem remains unclear: Do LLM agent groups exhibit a form of "general intelligence" that reflects their general ability across various tasks? In human cognitive psychology research, it has been established that the mental capabilities of a human group can be measured by a single statistical factor, known as the Collective Intelligence (CI) factor. This factor can capture the group's general capability and predict its performance on a wide range of tasks, much like how IQ scores capture the general cognitive ability of individuals. Inspired by this, in this study, we aim to investigate whether an analogous CI factor also exists in LLM agent groups, which is crucial for building generalizable MAS. Motivated by human cognitive psychology experiments, we design experiments along three dimensions: group size, individual intelligence, and collaboration process. Specifically, we construct 108 LLM agent groups with diverse group sizes, LLM compositions, and communication topologies. These groups are systematically evaluated across a wide range of tasks, including commonsense reasoning, math, game, etc. Our results demonstrate that an Artificial Collective Intelligence (ACI) factor does exist in LLM agent groups, accounting for 66.3% of the variance in performance across different tasks, which is substantially higher compared with the 43% observed in human groups. Moreover, by analyzing the indicators of groups that affect ACI, we find similar patterns between the ACI of LLM agent and human groups, where the collaboration process is the most important indicator influencing ACI rather than the individual intelligence of group members. This highlights that, for MAS design, the way agents are connected and interact has a greater impact on overall performance than the scale of individual models, offering practical guidance for building more efficient and generalizable MASs. Our code is open-source at `https://anonymous.4open.science/r/LLM_Collective_Intelligence-71B3` for reproducibility.

## 1 Introduction

The rapid development of large language models (LLMs) has given rise to LLM-based multi-agent systems (MAS), which have shown remarkable capabilities in many domains. Prior studies reveal that different MAS may excel in different tasks (Zhang et al., 2024b), and thus researchers have proposed a variety of methods to design MAS optimized for specific applications, such as coding (Qian et al., 2024a) and game playing (Chen et al., 2023). However, a fundamental question remains unclear: do LLM-based MAS exhibit a form of "general intelligence" that goes beyond task-specific performance and reflects a group's overall ability across diverse tasks?

In human cognitive psychology research, the quest for a "general intelligence" measure has a long history (Spearman, 1904), with the most popular test known as the "IQ test". This line of research seeks to derive a single statistical factor that measures the generalizable mental capabilities of individuals across various cognitive tasks. More recently, studies have shown that the cognitive performance of human groups can also be predicted to a large extent by a single statistical factor,

which is referred to as the "collective intelligence" (CI) factor (Woolley et al., 2010; Riedl et al., 2021). This factor captures the task-independent capability of groups across a wide range of domains. Since LLMs have shown many human-like behaviors (Chen et al., 2025), a natural question is whether a similar CI factor also exists in LLM agent groups. If so, it not only indicates that LLM agent groups share similar general intelligence with human groups, but also would provide critical insights for designing more effective and generalizable LLM agent groups.

In this work, we conduct systematic experiments to investigate the existence and properties of the CI factor of LLM agent groups. We aim to answer three research questions: (1) Does a general CI factor exist in LLM agent groups? (2) What are the most important indicators of LLM agent groups that affect CI? (3) Can insights from CI be used to guide the design of LLM agent groups? To answer these questions, we construct 108 LLM agent groups spanning 8 different LLMs, while varying group size, communication topology, and model composition. These dimensions are chosen based on human experiments (Riedl et al., 2021), which ensure the diversity and robustness of our experiments. We then evaluate the groups on a broad spectrum of cognitive tasks, including commonsense reasoning, mathematics, game playing, coding, and writing. Our findings can be summarized as follows. First, we provide evidence for the existence of a general CI factor, which we term Artificial Collective Intelligence (ACI), in LLM agent groups, which captures group ability and generalizes across tasks. Second, ACI in LLM agent groups shows similar patterns with CI in human groups, where the collaboration process is the most important determinant of ACI, outweighing the individual intelligence of group members. This suggests that it is possible to design lower-cost yet high-performing MASs; for example, our case study shows that an alternative design can reduce cost by 43% while improving ACI by 9.7%. Third, we show that the indicators of LLM agent groups can be used to predict the performance of new groups, offering a practical pathway to optimize group design at lower cost. The main contributions of the present work are threefold:

- We demonstrate the existence of a general ACI factor in LLM agent groups, which accounts for 66.3% of the variance in group performance and generalizes well across tasks.

- We analyze the indicators of LLM agent groups that affect the ACI factor and find similar patterns with human groups. Specifically, the collaboration process has the greatest impact on ACI, followed by individual intelligence, with group size having a relatively smaller effect. Moreover, we show that these indicators can be used to predict the performance of LLM agent groups.

- Based on these findings, we propose practical design principles for LLM agent groups, such as putting stronger agents on high-degree nodes within the communication networks.

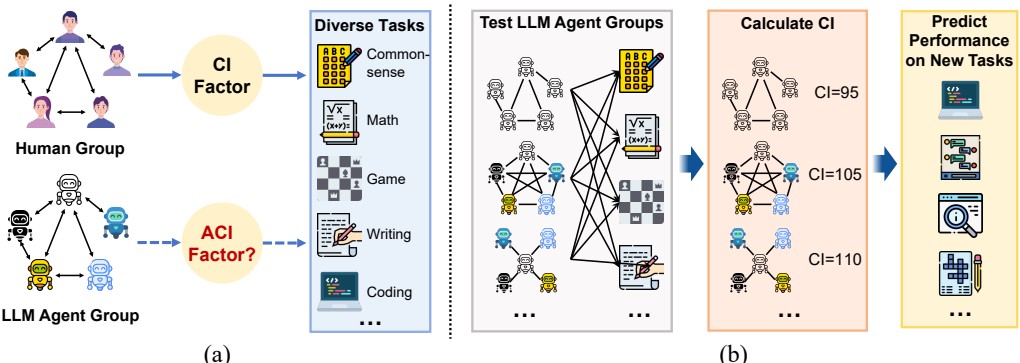

Figure 1: (a) We aim to investigate whether LLM agent groups also exhibit an ACI factor similar to that observed in human groups. (b) The overall framework of our experiments.

## 2 RELATED WORK

### 2.1 COLLECTIVE INTELLIGENCE OF HUMAN

Individual intelligence of humans is commonly conceptualized as a statistical factor, which predicts performance across various tasks (Spearman, 1904). Similarly, CI describes a group's ability to perform a range of tasks, also captured by a single statistical factor. Woolley et al. demonstrated the existence of CI factor in human groups, which accounts for over 40% of the variance in group

performance (Woolley et al., 2010). They also found that CI is correlated not only with the individual intelligence of group members but also with their average social sensitivity and the proportion of females in the group. Riedl et al. conducted large-scale experiments with more than one thousand groups, which further verified the existence of CI (Riedl et al., 2021). They also found that the group collaboration process is more important in predicting CI than individual intelligence. These studies on CI in human groups provide a valuable framework for investigating the CI in LLM-based multi-agent systems. LLMs have demonstrated many human-like behaviors, and it has been pointed out that individual LLMs show interrelated cognitive-like capabilities like humans (Ilić & Gignac, 2024). However, it remains unclear whether groups of LLM agents also have a general CI factor.

## 2.2 LLM Multi-agent Collaboration

LLMs have demonstrated outstanding role-play and reasoning ability, which enables them to collaborate with other LLM agents to solve complex tasks (Xiao et al., 2023; Li et al., 2023; Hong et al., 2023; Qian et al., 2024a; Chen et al., 2023). In recent years, there have been extensive studies on multi-agent collaboration, which can be categorized into three types. The first line of studies aims to design multi-agent collaboration methods for specific tasks, usually based on human collaboration mechanisms. For example, Du et al. design a multi-agent debate framework, where multiple agents debate for several rounds to solve a problem (Du et al., 2024). MetaGPT follows the standardized operating procedures in human software development process and proposes a multi-agent collaboration framework for software development (Hong et al., 2023). Another line of studies further proposes to automatically design and optimize the collaboration strategy. For instance, Agentverse lets LLM generate and adjust the agent composition based on the status of the task (Chen et al., 2023). G-designer proposes to optimize the communication network of agents through a variational graph auto-encoder (Zhang et al., 2024b). GPTSwarm represents multi-agent systems as composite graphs and optimizes node-level prompts as well as edges between agents (Zhuge et al.). Moreover, a third line of studies focuses on the underlying mechanism of multi-agent collaboration, such as the impact of agents' traits (Zhang et al., 2024c) and hyperparameters (Smit et al., 2024), and the scaling law of multi-agent systems (Qian et al., 2024b). However, existing studies mainly focus on task-specific scores and overlook the general ability of LLM agent groups across diverse tasks.

## 3 Experiment Framework

In this study, we investigate the CI of LLM agent groups from the following aspects:

**1. Does an ACI factor exist in LLM agent groups?** We conduct factor analysis to extract the latent factor from the performance of different LLM agent groups across a wide range of tasks, which shows that there exists a factor accounting for 66.3% of the variance. (Section 4)

**2. What are the most important indicators of LLM agent groups that affect ACI?** We analyze the characteristics of LLM agent groups that affect their ACIs, and find that it is the collaboration process that influences ACI most. (Section 5.1)

**3. Can insights from ACI be used to guide the design of LLM agent groups?** We demonstrate that the features of LLM agent groups can be used to predict ACI for unseen groups, which could help estimate the group performance without testing on specific tasks. (Section 5.2 and 6.1)

We first introduce our experiment framework as follows.

## 3.1 Multi-agent Collaboration Framework

We leverage a widely used LLM multi-agent collaboration framework (Du et al., 2024; Wang et al., 2025; Yu et al., 2024), where multiple LLM agents discuss for several rounds to answer a question. Specifically, the LLM agents can be modeled as a graph $\mathbf{G} = \{\mathcal{V}, \mathcal{E}\}$, where $\mathcal{V} = \{v_1, v_2, \ldots, v_N\}$ is the set of nodes, each node is an LLM agent, and $\mathcal{E}$ is the set of edges. We also refer to the graph $\mathbf{G}$ as the *communication topology* of LLM agent groups. Given a query $q$, each agent $v_i \in \mathcal{V}$ independently generates an initial response $r_i^{(1)} = v_i(q)$. Then in round $t(t \geq 2)$, each agent observes the previous answers of neighboring agents, and updates its own answer:

$$r_i^{(t+1)} = v_i(\{r_j^{(t)} | j \in \mathcal{N}(v_i)\}), \tag{1}$$

where $\mathcal{N}(v_i)$ denotes the neighboring nodes of $v_i$. After $T$ rounds, the final answer is obtained by aggregating the responses of all agents

$$r^{(T)} = Aggregate(r_1^{(T)}, r_2^{(T)}, \ldots, r_N^{(T)}). \tag{2}$$

## 3.2 Composition of LLM Agent groups

We choose 8 different LLMs from various families to ensure diversity, including OpenAI (gpt-3.5-turbo-0125, gpt-4o-mini-2024-07-18), Qwen (Qwen2.5-7B-Instruct, Qwen2.5-32B-Instruct, Qwen2.5-72B-Instruct), GLM (glm-4-9b-chat), InternLM (internlm2_5-20b-chat), and Google (gemma-2-27b-it). Using these models, we construct LLM agent groups with varying group sizes, number of rounds, communication topologies, and LLM compositions. Specifically, the group sizes range from {3,5,8}, the number of rounds is set to {2, 3}, and the communication topologies include {decentralized Network, centralized network, random network}, which are described as follows.

- **Decentralized Network**: It is defined as a fully connected graph in which every pair of nodes is connected by a unique edge, i.e., each agent can receive the answers from all other agents.

- **Centralized Network**: It corresponds to a star graph structure where a central node is connected to all other nodes.

- **Random Network**: We generate random graphs using the Erdős–Rényi (ER) model (Erdős & Rényi, 1960) and Watts–Strogatz (WS) model (Watts & Strogatz, 1998). In the ER model, each pair of vertices is independently connected with a certain probability $p$. The WS model generates small-world networks by starting with a regular lattice and randomly rewiring edges with a certain probability.

Additionally, each group is composed of either homogeneous (same LLM) or heterogeneous (different LLMs) agents, resulting in a total of 108 groups. Their details are shown in Appendix A.1.

## 3.3 Datasets and Metrics

We evaluate the performance of LLM agent groups on five benchmarks: commonsense reasoning, mathematics, games, coding, and writing. The task selection covers widely adopted benchmarks in multi-agent system research (Zhuge et al.; Zhang et al., 2024b; Zhou et al., 2025), providing a diverse and representative set of tasks that effectively assess the collective intelligence of LLM agent groups.

- **Commonsense**: We choose the MMLU-Pro (Wang et al., 2024) benchmark, which is a more challenging version of MMLU (Hendrycks et al.) dataset containing multiple-choice questions with four to ten options. It contains problems from various disciplines, serving as a benchmark to test the general knowledge and commonsense reasoning ability of LLMs. The performance of LLM is measured by accuracy.

- **Math**: We use the MATH (Hendrycks et al.) benchmark, which contains math problems to test the mathematical reasoning ability of LLMs. The performance is measured by accuracy.

- **Game**: We use the Chess move validity tasks from BIG-Bench Benchmark (Srivastava et al., 2023), where the LLM agent is asked to provide a valid move of a piece given the history of chess moves. The performance is also measured by accuracy.

- **Coding**: We choose HumanEval (Chen et al., 2021), a widely used benchmark to measure the ability of function-level code generation. We use the *pass@1* metric to measure the correctness of generated functions on test cases.

- **Writing**: We use the Commongen-Hard (Madaan et al., 2024) benchmark. Each problem in this dataset consists of 20-30 concepts, and the agent is asked to generate coherent sentences that include all these concepts, which measures its reasoning and text generation ability. The performance is measured by the percentage of covered concepts (Chen et al., 2023).

More implementation details are presented in Appendix A.1.

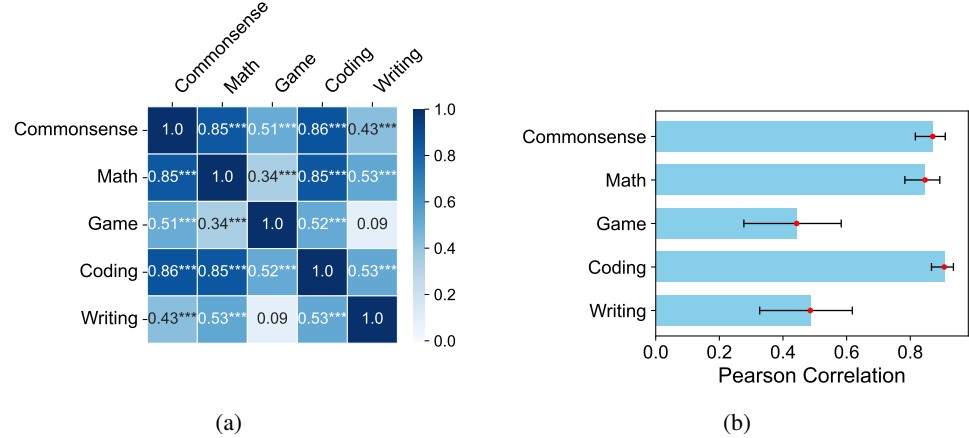

(a)                                                                 (b)

Figure 2: (a) Correlations between tasks. $***p < 0.001, **p < 0.01, *p < 0.05$. (b) Correlation of leave-one-out ACI with criterion task.

## 4 EVIDENCE FOR COLLECTIVE INTELLIGENCE FACTOR IN LLM AGENT GROUPS

### 4.1 EXISTENCE OF ACI

We first demonstrate that a general ACI factor exists in LLM agent groups. First, the performances of LLM agent groups across different tasks show a strong positive correlation, as shown in Figure 2(a). The average correlation coefficient is $r = 0.55$, notably higher than the $r = 0.28$ observed in human groups (Riedl et al., 2021). This strong cross-task correlation suggests the presence of a shared underlying capability—analogous to the general CI factor found in human groups—that influences group performance across different tasks.

To further examine this possibility, we perform exploratory factor analysis (EFA) to assess whether a single latent factor can account for performance variation across tasks. The analysis reveals a dominant factor that explains 66.3% of the total variance, substantially more than the 43% reported in human groups, while the second factor accounts for only 18.7%. We then conduct confirmatory factor analysis (CFA) by fitting a single-factor structural model. The resulting fit indices ($\chi^2 = 30.6$, $p < 0.001$, CFI = 0.967) indicate a good model fit, further supporting the presence of a general ACI factor. Taken together, these findings demonstrate that LLM agent groups, much like human groups, exhibit a form of collective intelligence that reflects a generalizable capability across tasks—one that appears even more pronounced than in human groups.

### 4.2 QUANTIFYING ACI

Based on previous analysis, we define an ACI factor of LLM agent groups following the definition of CI in human groups (Woolley et al., 2010; Riedl et al., 2021). Specifically, we first standardize the performance scores on each dataset because the scales of scores may vary across datasets. Let $s_{ij}$ be the standardized score of group $j$ on dataset $i$. Using the aforementioned factor analysis, we obtain a factor loading $w_i$ for each dataset $i$ (all $p < 0.001$), which reflects how strongly each observed variable (i.e., the performance on each dataset) is associated with the underlying ACI factor. Then the ACI factor of group $j$ is computed as the weighted score across all datasets

$$\text{ACI}_{j,raw} = \sum_{i=1}^{5} w_i s_{ij} / \sum_{i=1}^{5} w_i. \tag{3}$$

Following conventions in intelligence testing, we standardize these raw ACI scores by scaling them to have a mean of 100 and a standard deviation of 15:

$$\text{ACI}_j = \frac{\text{ACI}_{j,raw} - \text{mean}(\text{ACI}_{raw})}{\text{std}(\text{ACI}_{raw})} \times 15 + 100. \tag{4}$$

The resulting ACI scores for all LLM agent groups are reported in Appendix A.1.

To verify the generalizability of the ACI factor, we perform leave-one-out experiments where we use one of the five datasets as the held-out criterion task and compute the ACI factor using the remaining four datasets. We then assess how well these leave-one-out ACI scores predict group performance on the held-out task. As shown in Figure 2(b), the correlations exceed 0.8 on three of the tasks, and reach around 0.5 on the rest tasks, all statistically significant with $p < 0.001$. These results indicate that the ACI factor derived from any subset of four tasks generalizes well to unseen tasks, supporting its robustness as a measure of general group capability.

## 5    Patterns of ACI in LLM Agent Groups

### 5.1    Predictors of ACI

We have demonstrated that LLM agent groups have an ACI factor similar to human groups. An emerging question is what characteristics of a group affect its ACI the most?

Existing studies have shown that the CI of a human group is affected by indicators like group size, individual intelligence, and collaboration process (Woolley et al., 2010; Riedl et al., 2021). Following these findings, we construct a set of indicators for LLM agent groups with three categories as follows.

- **Group Size**: These indicators measure the size of a group, including the number of agents in a group ($N$) and its square ($N^2$).

- **Individual Intelligence**: These indicators characterize the ability of agents in a group. It has been demonstrated that individual LLM exhibits a general intelligence factor (Ilić & Gignac, 2024). Here we adopt the same method as calculating ACI (Section 4.2) to obtain an individual intelligence score $g$ for each LLM agent. We use the average $g$ and maximum $g$ of all agents in a group as indicators.

- **Collaboration Process**: These indicators describe how agents collaborate to solve the tasks (Hackman, 1978; Riedl et al., 2021). (1)*Variance of degree* is calculated as the variance of degrees of each node. It corresponds to the inequality of speaking turns in human groups, which has been demonstrated to be negatively correlated with CI (Woolley et al., 2010). (2)*Effort* is calculated as the total amount of activity that all agents perform during the task completion process. In our collaboration process, the activity refers to the communication between agents. Therefore, we define *Effort* as the number of rounds times the number of edges in the graph, i.e., *Effort*$= T \times |E|$ (3) *Skill congruence* measures the extent to which agents contribute efforts in proportion to their ability. In other words, a group where agents with higher capabilities put in more effort would have a high congruence. We define this indicator as the Pearson correlation between agents' individual intelligence and their node degrees. Experiments in human groups show that skill congruence is a strong positive predictor of CI.

It should be noted that we ignore some predictors in human groups that are hard to quantify in LLM agent groups, such as social perceptiveness (Baron-Cohen et al., 2001).

We present the standardized regression coefficient of these indicators predicting ACI in Figure 3(a). Consistent with human experiments, skill congruence and average individual intelligence are both significant positive predictors of ACI, while group size and effort are not strong predictors.

To assess the relative importance of each indicator, we fit a regression random forest model, which can capture nonlinear and more complex relationships between the indicators and ACI, and calculate the importance of each variable. As shown in Figure 3(b), the collaboration process plays the most significant role in predicting ACI, even more important than individual intelligence. We also fit a model to predict group performance on each of the datasets, yielding similar results. This finding aligns with prior research on human groups (Riedl et al., 2021). Specifically, the skill congruence indicator accounts for more than 50% of the total importance, and the average individual intelligence accounts for 37%. In comparison, the maximum individual intelligence, group size, and effort account for less than 5%. This is somewhat counterintuitive, as one might expect that a group's performance would be primarily determined by the individual abilities of its members. However, our findings suggest that the way agents interact with each other has a greater impact.

The implications are two-fold. First, simply increasing the ability of individual agents, such as employing stronger LLMs, does not necessarily lead to better outcomes. We present a case in

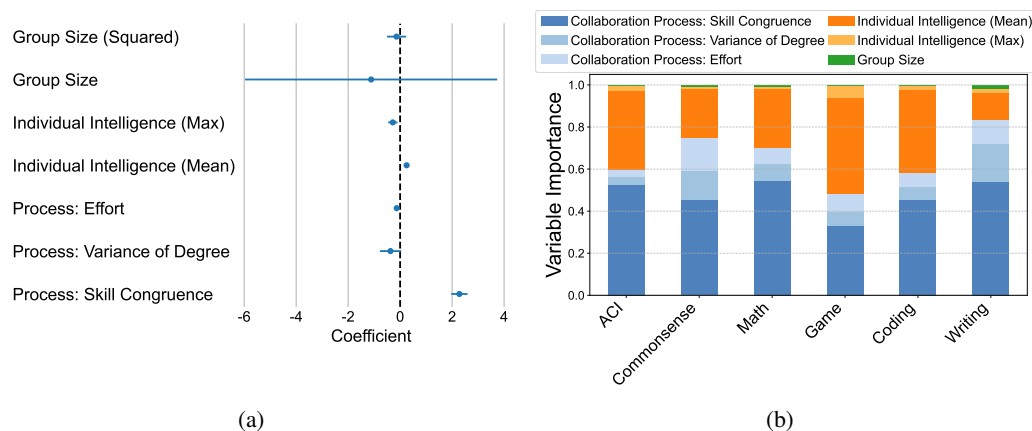

(a)                                            (b)

Figure 3: (a) Regression coefficients of indicators predicting ACI. (b) Importance of different indicators predicting ACI and task performances.

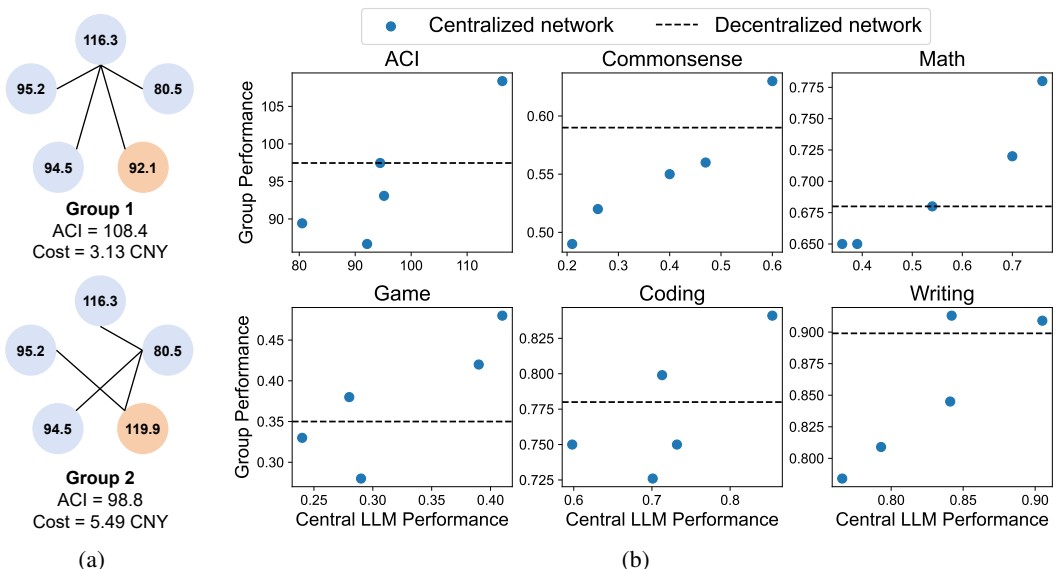

(a)                                            (b)

Figure 4: (a) Comparison of two LLM agent groups, where the second one has stronger LLMs and higher cost but lower ACI. The number in each circle represents the individual intelligence score $g$ of LLM. The cost is the total API cost for five tasks. (b) Comparison of decentralized networks and centralized networks with different LLMs serves as the central node. The red line shows the ACI/performance of the decentralized network. The blue dots show the relationship between the ACI/performance of the whole group and that of the central agent in centralized networks.

Figure 4(a), where the second group has a stronger LLM (Qwen2.5-72B, $g = 119.9$) than the first group (internlm2_5-20b, $g = 92.1$). Consequently, the second group also incurs a cost 75% higher than the first group. However, the ACI of the first group is 9.7% higher than the second one, highlighting the critical role of communication topology.

Second, compared with adding more communication links between agents, it would be better to let each agent do what matches their capabilities. In our collaboration framework, this means that stronger agents should be placed on nodes with higher degrees. We further verify this by comparing the performance of decentralized networks with centralized networks. Specifically, we construct six groups with five different LLMs (glm-4-9b-chat, internlm2_5-20b-chat, gemma-2-27b-it, Qwen2.5-7B-Instruct, and Qwen2.5-32B-Instruct), including a decentralized network and five centralized networks, where we let different agents serve as the central node. The ACI and performance of these groups on all datasets are presented in Figure 4(b). It can be observed that in centralized networks,

the task performances and ACI are positively correlated with those of the central agents, which is consistent with previous findings. Moreover, when the strongest LLM serves as the central node, the group performance not only achieves the best among centralized groups in most cases but also surpasses the decentralized network. Additionally, since the edges in a centralized network are a subset of the edges in a decentralized network, the centralized network has a lower time and token cost than the decentralized network. Such results suggest that with proper design of communication topology, the agent group can achieve better performance with lower cost.

## 5.2 Predicting the Performance of New Groups

We further examine whether the previously defined indicators that predict ACI can generalize to unseen groups. Specifically, we conduct a 2-fold cross-validation experiment, using half of the groups to fit a random forest regression model and predict the ACI and task performance of the rest groups based on the indicators. As shown in Figure 5, the $R^2$ achieves over 0.8 on ACI prediction, and over 0.6 on most of the tasks, suggesting a good generalization ability. Note that the indicators for LLM agent groups are solely dependent on the configuration of multi-agent collaboration, and there is no need to test the group on the target tasks. As a result, these indicators offer a promising way to predict the performance of new groups without incurring time or token costs.

Moreover, in many cases, such as when designing a multi-agent system, the goal is to identify the best groups instead of predicting exact performance. Therefore, we also present the mean reciprocal rank (MRR) metric for predicting the best group in Figure 5. The results indicate that the MRRs for ACI, Commonsense, and Game datasets exceed 0.35, meaning the best group is typically within the top-3 predicted groups. For the coding and writing tasks, the best group can be found within the top-6 predicted groups. On the Math dataset, the model can even achieve 100% accuracy in identifying the best LLM agent group. These findings highlight the potential of using these indicators to optimize the design of LLM multi-agent systems.

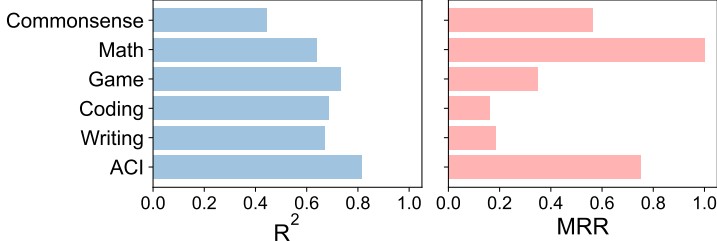

Figure 5: Results of predicting ACI and task performances using group indicators, evaluated by $R^2$ and mean reciprocal rank (MRR) for predicting the optimal group.

# 6 Discussion

## 6.1 Guidelines for LLM Agent Group Design

There have been studies on optimizing the configurations of LLM agent groups, such as prompt and topology, to improve their performance on certain tasks (Zhuge et al.; Zhang et al., 2024b;a). While these works are based on the assumption that the optimal group structure varies across different tasks, our study indicates that an LLM agent group has a general factor that characterizes its ability across tasks. This might seem contradictory at first glance, but the relationships between our study and these studies can be explained as follows. The ACI we find actually captures the *capability* (or *potential*) of a group instead of its *performance* on certain tasks. According to previous analysis, ACI captures both the individual ability and the alignment of individual abilities during the collaboration process, which is facilitated by group members' capacity to understand and interpret the intentions and goals of others (Veissière et al., 2020). This capability can predict the general task performance to some degree, while the performance is also affected by the characteristics of the specific task. This could somehow be demonstrated by the difference in the importance of collaboration process and individual intelligence (Figure 3(b)). For example, on writing task that requires divergent thinking and aggregation of ideas from different agents, the collaboration process contributes more to the

performance. On the contrary, performance on the game task with closed-ended questions is more affected by individual intelligence.

On the other hand, our findings can serve as a general principle to guide task-specific group structure optimization algorithms. For example, we find that it is generally better to place strong agents on nodes with higher degrees. However, this principle does not specify the exact topology of the group, as the optimal structure may still depend on the specific task, which can be found by optimization algorithms. Moreover, we demonstrate in Section 5.2 that we can predict the performance of groups based on some indicators as well as rank the best groups, which could be integrated into group optimization algorithms to make them more economical.

Overall, based on previous findings, we summarize the following guidelines for designing LLM agent groups:

- First, select high-performing LLMs. This is intuitive, and experimental results show that the individual intelligence of group members is a strong predictor of ACI.

- Second, align agents' efforts with their capabilities. This is supported by the finding that skill congruence is the most important predictor of ACI. In other words, assigning more capable agents to nodes with higher degrees will maximize their influence on the group, leading to better performance.

- Third, simply increasing the group size or effort does not yield significant benefits. Both the number of agents and the number of rounds have a minimal effect on ACI. Furthermore, creating a fully connected network among agents, as some previous studies suggest (Du et al., 2024; Estornell & Liu), is not necessary.

- Finally, it is possible to predict group performance and identify optimal configurations without conducting extensive experiments, thus reducing the cost of optimization algorithms.

## 6.2 LIMITATIONS

While this study provides an initial exploration of the ACI in LLM agent groups, several limitations must be noted. First, our findings are primarily based on empirical analysis rather than theoretical frameworks, which have limits on the understanding of the underlying mechanism of ACI. Second, regarding the multi-agent collaboration method, we focus on one typical multi-agent collaboration framework (Du et al., 2024). This limits the scope of our analysis, as other collaboration strategies, such as role-play or the assignment of distinct subtasks to different agents, were not considered. These alternative strategies may offer different insights into how ACI manifests in LLM agent groups. Finally, following the settings in human experiments (Riedl et al., 2021), we only consider groups with fewer than 10 agents. Although this scale is consistent with most of the existing LLM multi-agent collaboration frameworks (Qian et al., 2024a; Chen et al., 2023; Hong et al., 2023; Li et al., 2023; Du et al., 2024), the scalability of ACI in larger LLM agent groups remains an open question. It has been shown that under a certain collaboration framework, LLM agent groups exhibit a scaling law with a logistic growth pattern as the group size increases to one thousand (Qian et al., 2024b). Future exploration is needed to understand the pattern of ACI with larger group sizes.

## 7 CONCLUSION

In this study, we investigated the presence of an ACI factor in LLM agent groups, examining their general abilities across diverse tasks. Our extensive experiments revealed that LLM agent groups exhibit a generalizable ACI factor, accounting for 66.3% of the variance in performance, which can well predict the performance on other tasks. Furthermore, our analysis identified collaboration processes as the most critical determinant of ACI, rather than the individual intelligence of agents, mirroring patterns observed in human groups. This insight underscores the importance of designing effective collaboration strategies to enhance MAS performance and provide guidelines for MAS design. Finally, we demonstrated that key indicators of ACI can be leveraged to predict the performance of unseen groups, offering the potential for optimizing multi-agent collaboration with reduced computational costs. Our findings contribute to a deeper understanding of collective intelligence in LLM agent groups and pave the way for more efficient and generalizable MASs.

## 8 REPRODUCIBILITY STATEMENT

We provide the implementation details in Appendix A.1. The code and original data to reproduce results and figures in this paper are released at `https://anonymous.4open.science/r/LLM_Collective_Intelligence-71B3`.

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

# A  APPENDIX

## A.1  IMPLEMENTATION DETAILS

### A.1.1  PROMPT AND PARAMETERS

We use the prompts from the datasets' original papers for all tasks and adopt a zero-shot setting. To ensure the diversity of the agents' output, we set the temperature parameter to 1.0 for all experiments (Zhang et al., 2024b). We employ majority voting to aggregate the answers of all agents in a group. Specifically, for closed-ended questions (Commonsense, Math, Game), we calculate the most frequent answer. For open-ended questions (Coding, Writing), we follow a previous work (junyou li et al., 2024) and find the answer that is most similar to others, i.e.,

$$r^{(t)} = \arg\max_{r_i} \sum_{j=1, j\neq i}^{N} sim(r_i^{(t)}, r_j^{(t)}), \tag{5}$$

where $r_i^{(t)}$ is the response of agent $v_i$ at round $t$, and the similarity is calculated as BLEU score (Papineni et al., 2002).

### A.1.2  COMPUTER RESOURCES

All experiments are conducted on Windows 10 OS. The Python version is 3.10.12. We use LLM API provided by Azure OpenAI [1] (for OpenAI models) and SiliconFlow [2] (for non-OpenAI models). The factor analysis is implemented using Python package `factor_analyzer` [3]. The code and original data to calculate ACI and reproduce figures in this paper are released at https://anonymous.4open.science/r/LLM_Collective_Intelligence-71B3.

### A.1.3  DETAILS OF LLM AGENT GROUPS

We present the communication topologies and LLMs of all LLM agent groups here, including centralized networks (Figure 6), decentralized networks (Figure 7), and random networks (Figure 8). For each topology, there are two LLM agent groups with 2 rounds and 3 rounds. We also present the ACI of each LLM agent group in the figures.

---

[1] https://azure.microsoft.com/en-us/products/ai-services/openai-service
[2] https://siliconflow.cn/
[3] https://github.com/EducationalTestingService/factor_analyzer

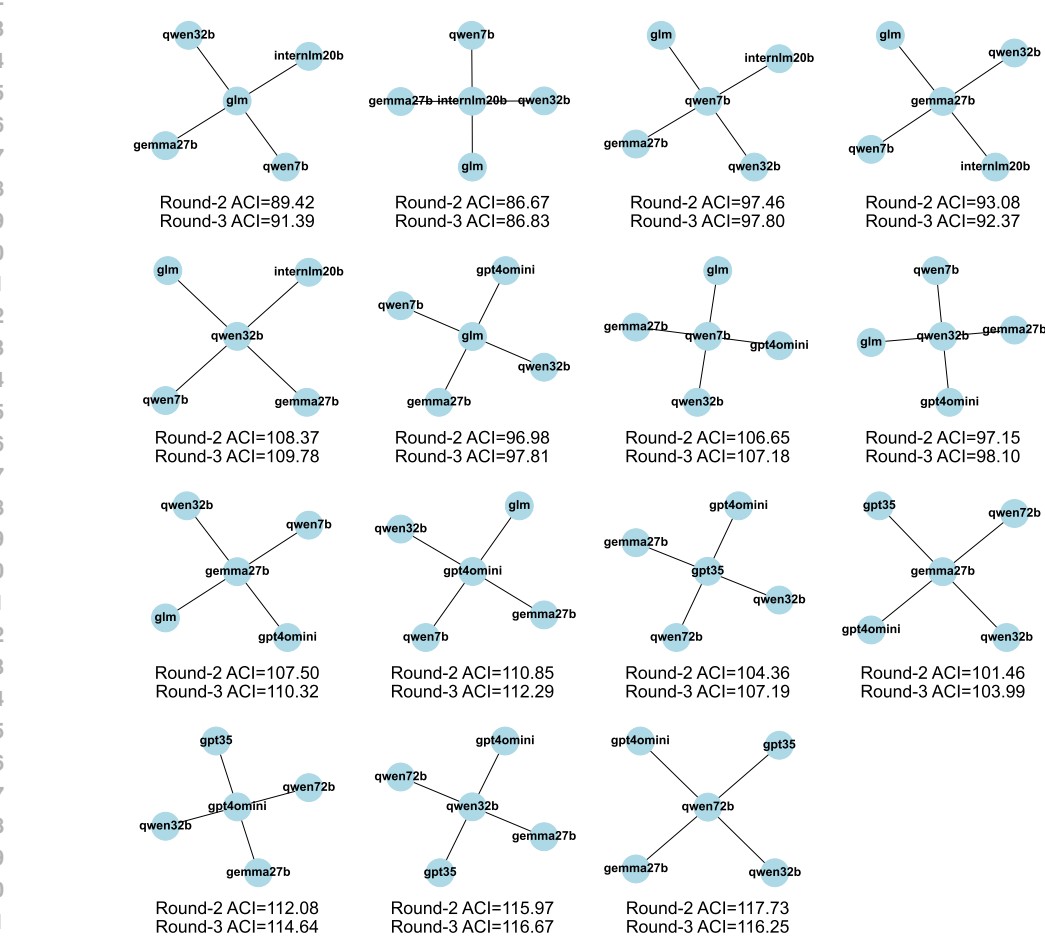

Figure 6: Topologies and ACIs of LLM agent groups with centralized network structure.

## A.2 FURTHER DISCUSSION

### A.2.1 CODE OF ETHICS

All datasets used in this study are publicly available, which involve no problem regarding privacy and copyright. No personally identifiable information was collected or used. We cite the resources in Section 3.3.

### A.2.2 BROADER IMPACTS

The implications of our findings are particularly significant for the development of Artificial General Intelligence (AGI). The emergence of a generalizable, task-independent ACI factor in LLM agents suggests that LLM agent groups possess an inherent mechanism that influences performance across various tasks. This mechanism could be related to factors such as agents' mutual understanding, shared cognitive processes, and the way they integrate their individual capabilities into a cohesive group effort. The presence of the ACI factor exhibits a form of *general intelligence* among the agents, which transcends specific tasks and contributes to their overall adaptability and effectiveness. Moreover, our findings point to the critical importance of collaboration in LLM agent groups. ACI in LLM agent groups demonstrates that, beyond individual capabilities, the way in which agents interact and collaborate can significantly affect their collective problem-solving abilities. This insight is foundational for advancing AGI, as it suggests that achieving human-like intelligence in artificial systems may depend less on replicating individual cognitive capabilities and more on fostering efficient collaboration within multi-agent frameworks. Finally, the ability of LLM agents to

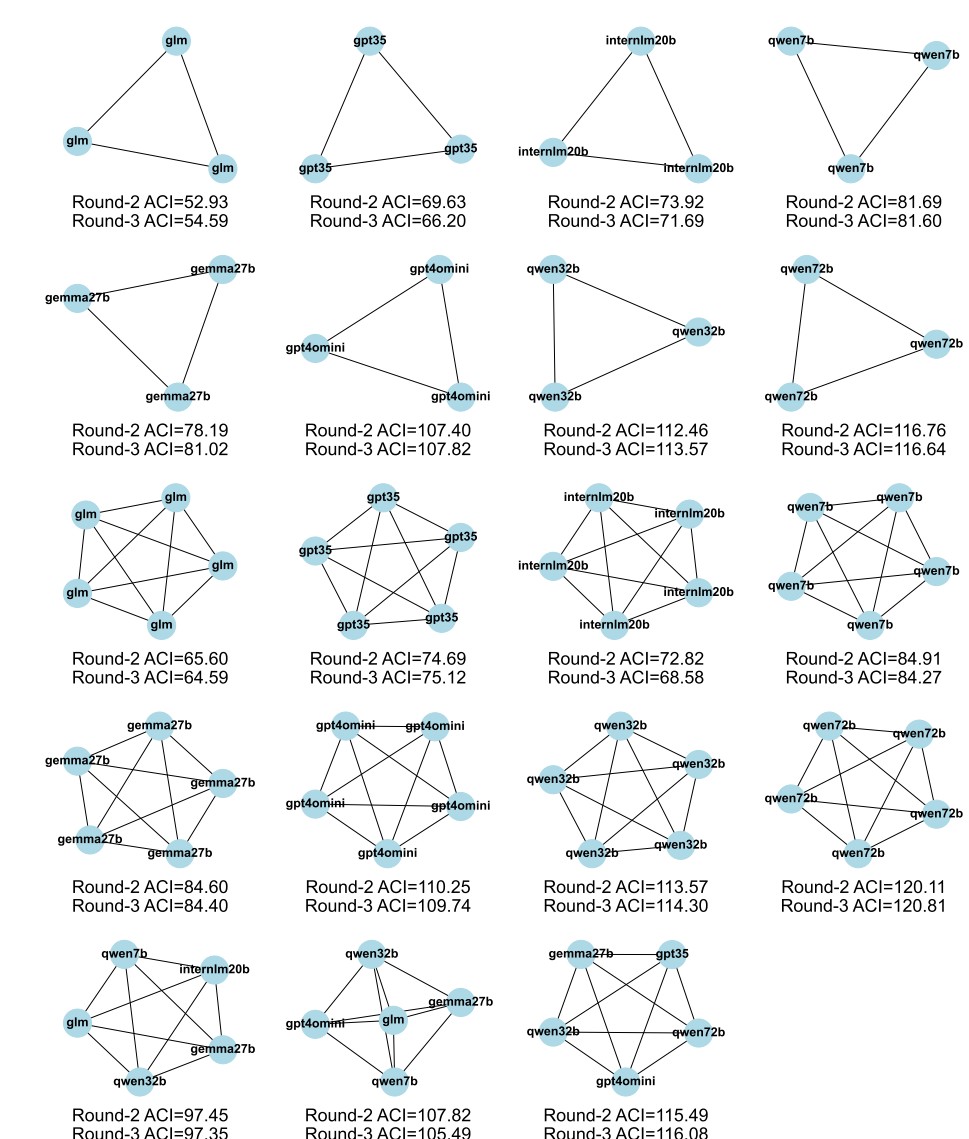

Figure 7: Topologies and ACIs of LLM agent groups with decentralized network structure.

exhibit a general intelligence factor, akin to human groups, also implies that scaling and optimizing these systems for increasingly complex tasks could follow a similar trajectory to human cognitive development, further accelerating the path toward AGI.

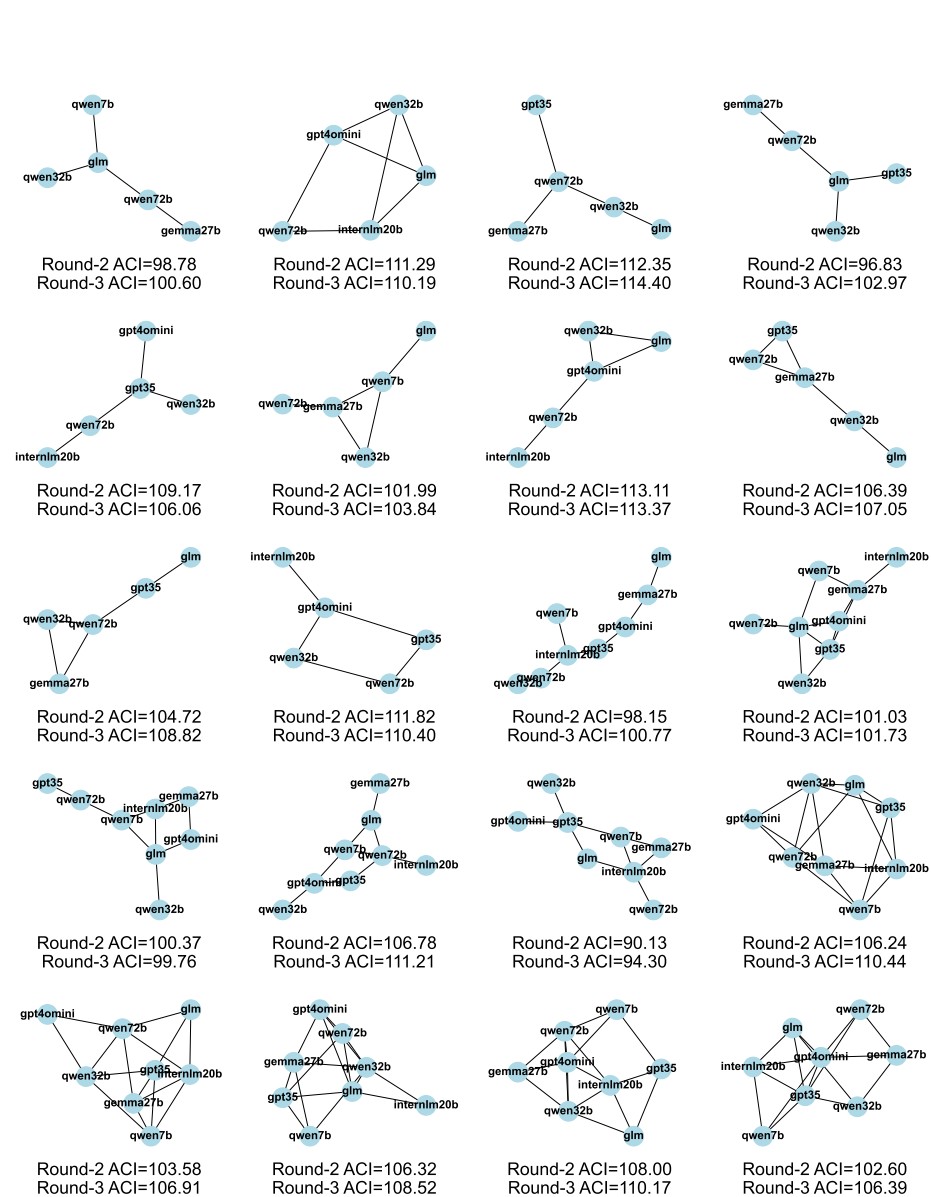

Figure 8: Topologies and ACIs of LLM agent groups with random network structure.

