# OpenReview forum: "Towards Generalizable LLM Multi-Agent System: Identifying Collective Intelligence Factor in LLM Agent Groups"
_ICLR.cc/2026/Conference — ICLR 2026 Conference Withdrawn Submission_

### Official Review · Reviewer_cD3W · 2025-10-21

**Soundness:** 2
**Presentation:** 2
**Contribution:** 1
**Rating:** 2
**Confidence:** 4

**Summary:**

The authors attempt to bring the concept of Collective intelligence into LLM agent group and answer the question: “Do LLM-based agent groups exhibit a form of general intelligence?”. The Collective intelligence derives from human cognitive psychology, which is proposed to measure the “capabilities of a group”. This work built several experiments to dig out potential CI factors in LLM agent group, from diverse aspects including group size, individual intelligence, and collaboration process, discovering that the skill congruence indicator is an essential CI factor to reflect the group intelligence.

**Strengths:**

-	This work proposes an interesting connection between LLM-based agent group and CI from human cognitive psychology.
-	Authors have conducted experiments and analyses on different tasks to find out the influential factors and patterns of ACI in agent groups.
-	This paper is well structured with open-source codes.

**Weaknesses:**

-	Adopting CI to rethink the patterns in LLM-based agent group is novel. However, the authors should explain what the innovative points and benefits of introducing CI are in the era of LLM-based agents? If introducing CI itself is the most essential contribution, the authors are also encouraged to provide a formal theoretical framework to define and analyze the pattern in LLM agent groups, rather than simply highlighting the connections between the two fields. A more quantified law is also beneficial for practical usage. The technical contribution should also be clarified.
-	Currently, the LLM-based agent-related tasks usually rely on much longer interactions. Moreover, whether the current tasks are challenging enough (for example, some tasks used in this work may be well solved by a single up-to-date flagship LLM. In this case, this task is not that suitable to explore multi-agent scenarios).
-	The “Skill congruence” is claimed to account for more than 50% of the total importance, even larger than individual intelligence. However, Skill congruence is more like a “weight” of different LLM agent individuals, which jointly reflects both collaboration process factors and individual intelligence factors. It is straightforward that the more usage of stronger LLMs, the better results we can achieve.
-	In my opinion, the conclusion of which indicator is the most essential for LLM agent group should depend on the difficulty level of the task over the LLM individuals. If the task is extremely challenging and we have one super LLM agent and four common agents, the individual intelligence indicator will be more significant. The super agent may not receive positive information from the agent group.

**Questions:**

See Weaknesses.

---

### Official Review · Reviewer_9Z9A · 2025-10-31

**Soundness:** 2
**Presentation:** 2
**Contribution:** 2
**Rating:** 4
**Confidence:** 4

**Summary:**

This paper explores whether groups of LLM agents exhibit a form of collective intelligence similar to that found in human groups. Drawing inspiration from cognitive psychology, the main motivation is whether an analogous Artificial Collective Intelligence (ACI) factor emerges in LLM-based multi-agent systems (MAS).

They construct 108 LLM agent groups spanning 8 model families, varying in group size, communication topology, and composition. Each group collaborates under a unified multi-agent framework across five cognitive tasks (commonsense reasoning, mathematics, games, coding, and writing).

Factor analysis reveals a dominant ACI factor explaining 66.3% of performance variance. A random forest analysis further shows that collaboration processes outweigh individual intelligence in determining overall performance. The proposed indicators generalize well to unseen groups, allowing performance prediction without additional task evaluation.

The study concludes that LLM agent groups exhibit a generalizable form of collective intelligence and offers practical design principles for building more efficient and scalable multi-agent systems.

**Strengths:**

1. The paper is built on a strong and well-defined motivation. The authors move beyond task-specific MAS optimization and ask a deeper question about whether LLM agent groups show a form of general intelligence across tasks. This system-level perspective gives the work clear theoretical depth and originality.

2. The empirical finding that interaction patterns and collaboration structures matter more than individual model scale is insightful. It shows that intelligence can be enhanced not only by stronger models but by better coordination among them.

3. The study builds 108 agent groups across 8 model families and systematically varies group size, communication topology, and model composition, making its conclusions robust.

4. The idea of applying the ACI framework to guide LLM group design is forward-looking. It suggests a way to optimize collective configurations without explicit task evaluation, turning collective intelligence from an observation into a principle for system design.

**Weaknesses:**

1. The claim that group size has little effect on collective intelligence is interesting but underdeveloped. Without further analysis, it remains a reported result rather than an explained phenomenon. Discussing potential causes such as cognitive load, coordination limits, or communication constraints would make the finding more convincing.

2. The paper draws parallels between human collective intelligence traits such as social sensitivity and gender composition and artificial collective intelligence, yet it does not explain how a single LLM can simulate different social roles across tasks. This conceptual gap weakens the foundation of the human–AI analogy.

3. Section 4 introduces many technical terms including exploratory factor analysis, latent factors, and factor loadings without clear definition or intuition. A concise methodological explanation would make the analysis more transparent and easier to follow.

4. The construction of the collaboration process metric is overly static and narrow. It is largely based on communication topology, which cannot by itself capture the richness of collaboration such as information exchange, role differentiation, or temporal coordination. This limitation weakens the claim that collaboration is the most important determinant of group performance.

5. The assertion that collaboration is the key driver of performance is not fully supported by the results. Among the defined collaboration metrics, only capability alignment shows a significant positive effect. This suggests that what matters may not be collaboration itself but whether high-performing agents are effectively leveraged. The positive correlation between group performance and central-agent intelligence in centralized structures further supports this alternative interpretation.

**Questions:**

-

---

### Official Review · Reviewer_jSVP · 2025-10-31

**Soundness:** 2
**Presentation:** 4
**Contribution:** 3
**Rating:** 4
**Confidence:** 4

**Summary:**

This paper explores whether an "artificial intelligence factor" (ACI) similar to that in human groups exists in LLM-based multi-agent systems. The authors designed 108 groups of agents composed of different LLMs, controlling variables from three dimensions: group size, individual intelligence, and collaborative process. Experiments were conducted on five tasks: commonsense reasoning, mathematics, games, programming, and writing. The results show that an ACI does exist, explaining 66.3% of the task performance variance. This ACI is primarily influenced by the collaborative process rather than individual model capabilities. Based on this, design principles for LLM-based multi-agent systems are proposed. The overall writing is clear, the experimental framework is complete, and the diagrams are intuitive, but the theoretical depth is slightly lacking.

**Strengths:**

1. This paper investigates an interesting question: whether human-like collective intelligence factors exist in LLM-based multi-agent systems, providing a new direction for the universality research of multi-agent systems.
2. 108 differentiated agent groups were constructed based on three core dimensions: group size, individual intelligence, and collaborative processes. These groups cover eight mainstream LLM systems, and experiments were conducted on five representative tasks. Multiple analytical methods, including EFA, CFA, and random forest regression, were used for cross-validation.
3. This paper provides valuable findings, suggesting that collaborative processes are more important than individual capabilities. It offers practical principles for designing low-cost, high-performance multi-agent systems (MAS), such as assigning highly capable agents to highly connected nodes and not needing to pursue fully connected networks excessively.
4. The open-source code ensures the reliability and reproducibility of the experiments.

**Weaknesses:**

1. The paper only considered a collaborative framework involving multiple rounds of discussion and did not include more complex MAS collaborative methods, such as role division and task decomposition.

2. The paper lacks a clear understanding and theoretical explanation for why ACI appears in LLM groups, remaining at the empirical level.

3. The paper found a high correlation between group performance on different tasks (writing, common sense reasoning, mathematics, programming, and games), thus inferring the existence of an ACI factor. However, this high correlation among the five tasks may stem from overlapping language abilities rather than genuine "cross-domain" intelligence factors. The authors' direct interpretation of "high correlation" as a logical chain for the existence of a group intelligence factor is questionable.

4. This paper's experiments only considered groups of 8 or fewer agents. However, existing research, such as "Scaling Large Language Model-based Multi-Agent Collaboration," has demonstrated that LLM-MAS performance exhibits a logistic scaling law as group size increases. This paper does not explore the variation of ACI in larger groups, and the conclusion that the collaboration process is more important cannot support the design requirements of large-scale LLM-MAS, making its application scenario highly effective.

5. The method for quantifying individual intelligence is questionable. This study uses task performance-based factor analysis to calculate the individual intelligence of a single LLM. Essentially, this only characterizes the consistency of LLM performance across tasks, which is fundamentally different from the innate cognitive abilities represented by human individual intelligence. Furthermore, it does not verify the correlation between this value and inherent technical attributes such as the number of LLM parameters, training data distribution, and pre-training objectives. This may lead to a disconnect between the individual intelligence index and the actual capabilities of the LLM, thus casting doubt on the accuracy and rigor of the subsequent conclusion that the collaboration process has a greater impact on ACI than individual intelligence.

**Questions:**

1. All five tasks selected in this paper were performed using natural language. Is it possible that ACI merely reflects the commonalities of language ability rather than true cross-task intelligence? Did you consider the statistical artifacts caused by semantic overlap between tasks? If non-linguistic tasks (such as image processing, action processing, or multimodal decision-making) were used, would the ACI factor still exist?

2. Why were these five tasks chosen for the experiment? Is there a sociological basis or theoretical support? The paper only lists the correlation analysis for these five tasks. Can the ACI factor be used to predict performance on unseen tasks?

3. Regarding the design principle of "placing strong agents in highly connected nodes," are there certain task types, such as those requiring divergent thinking, where this design principle might actually inhibit group performance?

4. Has the generalization ability of models or topologies completely unseen in the conclusions of this paper been tested? Does the model's predictive performance depend on the types of models already present in the training set? 5. In Figure 4(a), Group 1 has a higher ACI than Group 2. The authors attribute this to differences in communication topology. However, the LLM composition of the two groups (internlm2_5-20b vs Qwen2.5-72B) also differs. Has the interaction between LLM type and topology on ACI been ruled out? For example, has it been verified that replacing the LLM in Group 1 with Qwen2.5-72B still improves ACI?

6. The paper standardizes the performance of each task when calculating ACI. Did the difficulty differences of different tasks (e.g., MATH is more difficult than MMLU-Pro) take into account during the standardization process? If not, could this lead to an overestimation of the contribution of easier tasks to ACI, thus affecting the accuracy of ACI? If so, what specific methods were used to quantify task difficulty and correct the standardization results?

---

### Official Review · Reviewer_vDNw · 2025-11-01

**Soundness:** 3
**Presentation:** 3
**Contribution:** 3
**Rating:** 6
**Confidence:** 3

**Summary:**

This work shows that LLMs possess similar artificial collective intelligence(ACI) like human group. The authors also analyse indicators that affects the ACI and propose design principles to build stronger multi-agent networks.

**Strengths:**

- This work manages to measure intelligence of a multi-agent system with the cognitive psychology concept, collective intelligence
- The EFA, CFA strengthens the statistical grounding of the proposed ACI concept.
- The findings directly inform cost-efficient MAS design, emphasizing how topology and coordination matter more than model scale—useful for researchers and practitioners.

**Weaknesses:**

- This study is mostly empirical, with no formal theoretical framework to explain how ACI comes about or how it connects to LLMs' internal representations.
- While ACI captures shared variance, it might not fully stand for "general intelligence" in the cognitive sense. Correlation doesn’t always mean there’s a unified reasoning mechanism.

**Questions:**

- How sensitive are the ACI results to the chosen collaboration framework (discussion-based?）Would role-based or asynchronous collaboration yield similar ACI factors?

---

### Note · Authors · 2025-12-29

I have read and agree with the venue's withdrawal policy on behalf of myself and my co-authors.